# Contactless Temperature Sensing at the Microscale Based on Titanium Dioxide Raman Thermometry [note 1]

**DOI:** 10.3390/bios11040102

**Published:** 2021-04-02

**Authors:** Veronica Zani, Danilo Pedron, Roberto Pilot, Raffaella Signorini

**Affiliations:** 1Department of Chemical Science, University of Padua, Via Marzolo 1, I-35131 Padova, Italy; veronica.zani@studenti.unipd.it (V.Z.); danilo.pedron@unipd.it (D.P.); roberto.pilot@unipd.it (R.P.); 2Consorzio INSTM, Via G. Giusti 9, I-50121 Firenze, Italy

**Keywords:** temperature sensor, Raman spectroscopy, anti-Stokes/Stokes spectra, titanium dioxide

## Abstract

The determination of local temperature at the nanoscale is a key point to govern physical, chemical and biological processes, strongly influenced by temperature. Since a wide range of applications, from nanomedicine to nano- or micro-electronics, requires a precise determination of the local temperature, significant efforts have to be devoted to nanothermometry. The identification of efficient materials and the implementation of detection techniques are still a hot topic in nanothermometry. Many strategies have been already investigated and applied to real cases, but there is an urgent need to develop new protocols allowing for accurate and sensitive temperature determination. The focus of this work is the investigation of efficient optical thermometers, with potential applications in the biological field. Among the different optical techniques, Raman spectroscopy is currently emerging as a very interesting tool. Its main advantages rely on the possibility of carrying out non-destructive and non-contact measurements with high spatial resolution, reaching even the nanoscale. Temperature variations can be determined by following the changes in intensity, frequency position and width of one or more bands. Concerning the materials, Titanium dioxide has been chosen as Raman active material because of its intense cross-section and its biocompatibility, as already demonstrated in literature. Raman measurements have been performed on commercial anatase powder, with a crystallite dimension of hundreds of nm, using 488.0, 514.5, 568.2 and 647.1 nm excitation lines of the CW Ar^+^/Kr^+^ ion laser. The laser beam was focalized through a microscope on the sample, kept at defined temperature using a temperature controller, and the temperature was varied in the range of 283–323 K. The Stokes and anti-Stokes scattered light was analyzed through a triple monochromator and detected by a liquid nitrogen-cooled CCD camera. Raw data have been analyzed with Matlab, and Raman spectrum parameters—such as area, intensity, frequency position and width of the peak—have been calculated using a Lorentz fitting curve. Results obtained, calculating the anti-Stokes/Stokes area ratio, demonstrate that the Raman modes of anatase, in particular the E_g_ one at 143 cm^−1^, are excellent candidates for the local temperature detection in the visible range.

## 1. Introduction

The determination of temperature with good accuracy and with nano/micro-spatial resolution (nanothermometry) has been matter of intense research efforts since it opens up new perspectives in different research fields like biomedicine, photonics and nanoelectronics [1,2,3,4]. For example, Okabe et al. [5] reported on the investigation of cell functions by mapping the intracellular temperature, Quintanilla et al. [6] engineered a probe for monitoring temperature during photothermal therapy, Santos et al. [7] worked on the early diagnosis of tumors, exploiting the different thermal dynamics of healthy and diseased tissues and Mi et al. mapped the temperature on a micro sized magneto-resistive device [8].

In optical nanothermometry, luminescence is currently the most widely employed detection technique. However, also Raman spectroscopy is emerging as a valuable tool for temperature measurements. Despite being an intrinsically weaker phenomenon than fluorescence (requiring longer integration times), advantages of Raman include the wide range of temperatures detectable, the ease of sample preparation and the ample availability of materials possessing a Raman spectrum [9,10,11,12,13]. Moreover, it is characterized by a great spatial resolution, in the order of the diffraction limit of the probe laser [14].

The Raman effect is the inelastic scattering of light and well-defined characteristics of the Raman spectrum, such as intensity, position in frequency and width of peak signals, are related to temperature. It follows that temperature can be measured from Raman spectra by determining the degree of the shift position of a defined peak at different temperatures, or by evaluating the broadening of its linewidth or by measuring the peak intensity ratio of the anti-Stokes signal to the Stokes signal [15].

A good Raman thermometer material should possess these properties: (a) a large Raman scattering cross-section (to reach high signal-to-noise ratios); (b) high-intensity Raman peaks at low Raman shifts (the upper limit, in frequency shift, depends on the working temperature and in general near room temperature it is about 600 cm^−1^ [13]), indeed the lower the Raman shift, the more sensitive is the peak intensity to temperature; (c) well-defined and distinguishable Raman peaks and (d) low absorbance at the excitation wavelength (to avoid heating mechanisms).

Temperature cannot be measured directly, signals, like frequency position or intensity of the anti-Stokes and Stokes Raman peaks, can be used as indication (Q), which is linked to temperature through a mathematical equation (the so called measurement model). Great care should also be devoted to the determination of the uncertainty of the obtained values, indicating the dispersion of values within which the true temperature value is expected to lie. The measured temperature will be accurate if it is very close to the true value, and if measured temperatures, acquired by replicate measurements on the same object, are close to each other. In addition, sensitivity and thermal resolution are two fundamental parameters to evaluate thermometry. Sensitivity (S) is defined as the derivative of the indication with respect to the temperature, S=|∂Q/∂T| [9,16], while the thermal resolution is the smallest change in a temperature able to cause a perceptible change in the indication *Q*, calculated as the ratio of uncertainty (the standard deviation, σ) and sensitivity $\Delta T_{min}\;=\;\sigma /S$ [16].

When Raman thermometers are considered, it is also important to investigate the effect of the laser power on the local temperature, in order to avoid the heating of the sample due to the laser itself. For this purpose, it is interesting to examine the behavior of the system as a function of the laser intensity (when very high laser power intensities are achieved to obtain high intensity signals [17]). In general, the Raman signal depends on the third or the fourth power of the excitation frequency [18], so that it is expected to increase with decreasing excitation wavelength. That is the reason that many Raman measurements are conducted using shorter excitation wavelength; nonetheless, other factors have to be considered when choosing the proper laser frequency. Actually, it affects the depth of focus and the focal volume of the laser beam (the longer the wavelength the deeper light penetrates the sample), the spatial resolution and notably the photoluminescence background, which can be present also with transparent materials. Moreover, the presence of electronic transitions close to the excitation frequency has to be considered, as it causes an enormous enhancement of Raman signals [19], which is desirable to increase sensitivity, but may induce a local heating of the sample. When temperature is measured through the anti-Stokes/Stokes ratio, all these aspects have to be considered when choosing the material and/or the excitation wavelength, because a wavelength dependence of the Raman cross-section can determine an asymmetry between Stokes and anti-Stokes processes, resulting in anomalous anti-Stokes/Stokes ratios.

Studies on temperature with Raman measurements can be found in literature for silicon [14,20], gallium arsenide [21], gallium nitride [22] and graphene [23]. Titanium dioxide (TiO_2_) has also been tested in few works as a Raman thermometer for titanium dioxide microparticles [24,25] and for thin films of titania used in solar cells [26].

Titanium dioxide’s general features of chemical stability and nontoxicity make it a very interesting compound for various different applications, including photocatalysis [27], optical coatings, optoelectronic devices [28] and biomedicine [29]. It is a wide band gap insulator (3.0 eV [30,31,32]) and exists in nature in three different crystal structures: anatase, rutile and brookite. In particular, the anatase phase is exploited in photocatalysis, photochemical solar cells, optoelectronic devices and chemical sensors [33,34]. Titanium dioxide seems to fit perfectly all the requirements for a good thermometer material and has been chosen as Raman active thermometric material in our research.

The aim of this study is to obtain a protocol for temperature determination, with a high spatial resolution, of the order of the micro-nanometer dimension, exploiting Raman spectroscopy on anatase powder. As multiple signals are present in the Raman spectrum of Titanium dioxide, the choice of the actual Raman mode to be used has been performed on the base of its sensitivity to temperature. The ratio between Stokes and anti-Stokes signals of the same Raman mode has been investigated as a function of temperature (T), excitation wavelength ($\lambda_{exc}$) and input power. The control of the temperature is obtained by using a temperature controller, which is assumed also as reference for the determination of the absolute temperature. The performances of the temperature sensor are examined in the wavelength range 488.0–647.1 nm, to individuate the best excitation wavelength in terms of reaching the highest sensitivity, and in the temperature range 283–323 K, which is important for biological applications. The work will demonstrate that a different calibration constant is necessary for different wavelengths and Raman modes. The calibration constants, determined with this work, have been tested on a titanium dioxide based *Test Sample*, obtaining results with high sensitivity and low uncertainty and open the way to the use, in the future, of titanium dioxide-based new biosensors.

## 2. Materials and Methods

Raman measurements have been performed using a micro-Raman setup in a back-scattering geometry; the principal elements of the setup are showed in Figure 1.

The system is equipped with a CW Ti:Sapphire Laser, tunable in the range 675–1000 nm (MKS Instruments, Spectra Physics, 3900S, Santa Clara, CA, USA) and pumped by a CW Optically Pumped Semiconductor Laser (Coherent, Verdi G7, Santa Clara, CA, USA), and an Ar^+^/Kr^+^ gas laser (Coherent, Innova 70, Santa Clara, CA, USA) providing the lines at 488.0, 514.5, 530.8, 568.2 and 647.1 nm. The laser beam is coupled to a microscope (Olympus BX 40, Tokyo, Japan) and focused on the sample by 100×, 50× or 20× objectives (Olympus SLMPL, Tokyo, Japan). The Raman scattering is collected into the slit of a three-stages subtractive spectrograph (Jobin Yvon S3000, Horiba, Kyoto, Japan) by means of a set of achromatic lenses. The spectrograph is made up of a double monochromator (Jobin Yvon, DHR 320, Horiba, Kyoto, Japan), working as a tunable filter rejecting elastic scattering, and a spectrograph (Jobin Yvon, HR 640, Horiba, Kyoto, Japan). The Raman signal is detected by a liquid nitrogen-cooled CCD (Jobin Yvon, Symphony 1024 × 256 pixels front illuminated). When an entrance slit of 50 μm is used, a precision of 0.6 cm^−1^ in the determination of the peak position is obtained, with this experimental set-up.

A temperature-controlled stage (Linkam, THMS600/720, Tadworth, UK) is used to change and control the temperature of the sample, by means of a liquid nitrogen reservoir and heating resistances, giving a control of 0.1 K on the temperature inside the sample chamber. The sample is inserted into the temperature controller stage, and uniformly heated or cooled to reach the desired temperature, with a rate of 5 K/min and a thermalization time of at least 30 min. Before starting the experiment, a procedure of purging air from the stage chamber with nitrogen is performed; by this way the air in the chamber is eliminated and an inert static nitrogen atmosphere is realized, allowing fast temperature variations. Once the thermalization process is done, consecutive Stokes and anti-Stokes measurements are conducted to measure the local temperature of the sample. The wavelength incident on the sample is properly chosen in order to avoid sample heating (by using photons less energetic than the band gap of sample). Raman spectra have been collected in the visible range, by exciting at 488.0, 514.5, 568.2 and 647.1 nm, at different temperatures, ranging from 283 to 323 K. Measurements are repeated, at each wavelength and temperature, to obtain a consistent set of data (from 5 to 10 measurements at each temperature), by collecting the Raman signal in different positions of the sample.

Measurements of the anti-Stokes/Stokes ratio have been collected also at different laser powers, in the range 0.1–20 mW, to individuate the region where the signal is independent from the power, and the local temperature is not influenced by the presence of a laser beam. An input power of few mW has been used for temperature measurements.

All Raman spectra have been collected using a 20× objective (Olympus) with numerical aperture (N.A.) of 0.4 and a working distance of 12 mm, under these conditions the spot of the laser on the sample is expected to be approximately 1.5 μm (nearly equal for all the lines of the Ar^+^/Kr^+^ laser).

The sample is titanium dioxide, a commercial anatase powder (Sigma Aldrich, Merck KGaA, St. Louis, MO, USA), with a crystallite dimensions of ~200 nm; it possesses a band gap of 3.4 eV [32]. Titanium dioxide has been used as pristine powder inserted in the temperature stage and as powder pressed on KBr pellet sample (herein called *Test Sample*), with a final thickness of few hundreds μm.

## 3. Results

### 3.1. TiO_2_ Raman Spectra at 488.0 nm

The Stokes Raman spectrum of anatase powder, recorded at room temperature, using a laser at 488.0 nm with an input power of 1.72 mW, is reported in Figure 2. The spectrum clearly shows an intense peak centered at 143 cm^−1^ and four peaks, at 197, 397, 515 and 640 cm^−1^, with lower intensity.

Anatase crystals are characterized by 15 optical modes at the Γ point of the Brillouin zone, described with the following irreducible representation of the normal vibrational modes [28,35]:$$1A_{1g}+1A_{2u}+2B_{1g}+1B_{2u}+3E_{g}+2E_{u}$$

Among these, only six modes, $A_{1g}$, $2B_{1g}$ and $3E_{g}$ are the Raman-active ones (reported in Figure 2 and in Table 1); the Raman spectrum shows only five peaks since the *B*_1*g*_ and *A*_1*g*_ modes, at 512 and 518 cm^−1^ respectively, are not distinguishable at the experimental conditions, due to their intrinsic amplitude [28,36].

The experimental frequencies, reported in the central column of Table 1, corrected by using the cyclohexane frequencies as calibration frequency, are comparable to data reported in literature [28].

The Stokes (positive Raman shift) and anti-Stokes (negative Raman shift) Raman spectra are reported in Figure 3, where it is possible to observe also the zooms of the low intensity peaks. By observing Stokes and anti-Stokes data, it turned out that the best peak for temperature monitoring is the E_g_ mode at 143 cm^−1^. It is well defined, very intense even at low laser powers and highly sensitive to temperature, as will be demonstrated later, thanks to its low Raman shift.

### 3.2. Raman Spectra of TiO_2_ as A Function of Temperature, Excitation Wavelength and Input Power

Stokes and anti-Stokes Raman spectra of anatase have been collected in the temperature range of 283–323 K, by exciting at 488.0, 514.5, 568.2 and 647.1 nm, using an input power of 1.47, 1.20, 2.20 and 5.86 mW, respectively. The Raman spectra collected at room temperature, at different excitation wavelengths, are reported in Figure 4a, while the 143 cm^−1^ E_g_ mode, excited at 514.5 nm, at different temperatures, is illustrated in Figure 4b.

All Raman spectra have been analyzed with Matlab, using a Lorentz fitting, to obtain the Raman spectrum parameters, such as frequency position, width, intensity and area of the peak (see Appendix A).

An example of the data, obtained from the analysis of the Raman spectra centered on the 143 cm^−1^ peak, by exciting at 514.5 nm at different temperatures, is reported in Table 2, where data from the Stokes spectra are reported together with the area of the anti-Stokes peaks, which allows to calculate the anti-Stokes/Stokes ratio, a parameter important for the determination of the local temperature. The errors of the peak positions and widths are experimental errors, directly derived from the characteristics of the experimental set-up; the errors on the area have been derived from the fitting (see the example in Appendix A); for the anti-Stokes/Stokes ratio data, the error propagation has been considered [12].

Corresponding to the increasing in temperature, both the peak frequency position and the anti-Stokes/Stokes ratio clearly show an increment, from 142.5 to 145 cm^−1^ and 0.48 and 0.54, respectively, while the peak width varies in between 10.7 and 11.4 cm^−1^.

In order to test whether the Raman spectra, and the corresponding parameters, are perturbed by the laser irradiation, it is also necessary to investigate the effect of increasing incident laser power by keeping constant all other variables, like excitation wavelength. The laser power was changed in a range between 0.1 and 18 mW (depending on the laser wavelength); the experimental results (peak position, peak width (FWHM, i.e., full width half maximum), peak area and anti-Stokes/ Stokes ratio), for the 143 cm^−1^ E_g_ mode, at 514.5 nm, are shown in Figure 5.

It is evident that parameters are not increasing with the laser power only in the small range, 0–2 mW, while at higher input power they increase with the increase of the laser power. In order to assume that the temperature of the sample is not influenced by the presence of the laser irradiating the sample, it is necessary to keep the laser power at values lower than 2 mW.

### 3.3. Test Sample Measurement

A validation of the method can be done by evaluating anti-Stokes and Stokes intensities in various positions of the *Test Sample*, thus verifying if the parameters measured all over the sample are uniform or not, and comparable with the results previously obtained. The calculated anti-Stokes/Stokes ratios are reported in Table 3.

## 4. Discussion

Raman spectroscopy allows for temperature measurements with two methods: (a) Since the anti-Stokes and Stokes Raman intensities are proportional to the populations of their respective initial vibrational states, described by the Boltzmann distribution, the T of the sample can be estimated from the ratio of the Stokes and anti-Stokes intensities [37,38,39]; (b) the frequency position, the intensity or the shape of a (Stokes) Raman band may change as a function of temperature: The increase in temperature is expected to loosen chemical bonds (and hence to decrease the frequency of the mode), and/or induce more significant structural changes in the material under investigation [40,41,42]. Within this work, the local temperature has been investigated considering Stokes and anti-Stokes Raman spectra.

The anti-Stokes/Stokes ratio allows to determine the local temperature T of the sample, through the relation:(1)$$\rho =\frac{I_{aS}}{I_{S}}=C\cdot \frac{{\left(\nu_{0}+\nu_{m}\right)}^{3}}{{\left(\nu_{0}-\nu_{m}\right)}^{3}}\exp\left(-\frac{h\nu_{m}}{k_{B}T}\right)$$
where $\nu_{m}$ is the frequency of the vibrational mode m considered, $\nu_{0}$ the laser frequency, h the Planck’s constant, $k_{B}$ the Boltzmann constant and C is the calibration constant. A frequency dependence to the third power of the anti-Stokes/Stokes ratio is needed as the detection system is based on photon counting (CCD), whereas if the detection is energy-based, a fourth power dependence is more appropriate [40]. The calibration constant is related to the experimental setup, in particular to parameters such as the polarization of the incident laser and the CCD and grating efficiency. The determination of the calibration constant, at each working excitation wavelength, is a key point to the determination of the local temperature.

### 4.1. Determination of the Calibration Constant

Raw Raman spectra were analyzed to obtain the parameters, such as area, intensity, frequency position and width of the peak. In particular, the anti-Stokes area of the E_g_ mode was divided by that of the Stokes signal to obtain the experimental anti-Stokes/Stokes ratio, ρ (reported as example in Table 2—Section 3.2 for the 514.5 nm excitation wavelength).

With the Raman scattering cross-section being nearly constant over the wavelength range explored, normalized intensities still differ depending on the CCD and grating efficiency (instrumental response function), which is particular of the instrumentation set-up used.

Equation (1) was fitted to the experimental values, collected at different temperatures and defined excitation wavelength, leaving C as a free parameter. 

Figure 6 reports the anti-Stokes/Stokes ratio obtained from the calculated area of the anatase Raman modes, at 143, 397, 515 and 640 cm^−1^, in the temperature range 283–323 K, excited at 514.5 nm, and the curve resulting from the fitting with Equation (1) (the dashed lines). The corresponding calibration constants are calculated to be 0.9605, 0.9411, 0.9461 and 0.9491, respectively. 

The anti-Stokes/Stokes experimental ratios of the 143 cm^−1^ Raman mode, measured at different excitation wavelengths as a function of the temperature imposed by the thermostat are reported in Table 4 and depicted in Figure 7, together with the curve obtained from the fitting and the corresponding calculated calibration constants, reported in Table 5. The curve is well fitted to experimental data, as the R2 parameters range from a minimum of 0.84, at 647 nm, to a maximum of 0.95, at 514.5 nm.

As we can see qualitatively in Figure 6 and Figure 7 and quantitatively in Table 4 and Table 5, the calibration constant C is different depending on the excitation wavelength and the frequency of the Raman mode, due to the difference in response of the optical components of the experimental set-up to the wavelength. For this reason, it is necessary to individuate a calibration constant for a defined Raman mode, at a specific excitation wavelength.

### 4.2. Comparison between the Four Raman Modes of Titanium Dioxide

In order to individuate the best Raman mode, to obtain the more efficient signal in the determination of the local temperature, it is necessary to compare the behavior of the four TiO_2_ Raman modes as a function of the temperature variation. Figure 8a shows, for example, the theoretical anti-Stokes/Stokes ratios of the Titania modes, calculated with excitation wavelength at 514.5 nm, in the temperature range of interest, 283–323 K. The 143 cm^−1^ E_g_ anti-Stokes/Stokes ratio shows the highest value, in comparison with the other Raman modes, it presents values in the range 0.50–0.55, while the 397 cm^−1^ mode, the 515 cm^−1^ and the 640 cm^−1^ modes present values in the ranges 0.15–0.19, 0.08–0.12 and 0.05–0.07, respectively. The total variations of the ratio, in the whole T range (∆ρ), decreases from 0.0476, with the 143 cm^−1^ mode, to 0.0233, with the 640 cm^−1^ mode. At the excitation wavelengths of 488.0, 568.2 and 647.1 nm the behavior and the values obtained are comparable to that obtained at 514.5 nm.

To evaluate the efficiency in temperature detection, one of the most used figures of merit of thermometry is the sensitivity, calculated, for Raman measurements, through the following Equation [9,16]:(2)$$S=\left|\frac{\partial \rho }{\partial T}\right|=\left|-\frac{{\left(\nu_{0}+\nu_{m}\right)}^{3}}{{\left(\nu_{0}-\nu_{m}\right)}^{3}}\frac{h\nu_{m}}{kT^{2}}\exp\left(-\frac{h\nu_{m}}{kT}\right)\right|$$
where ρ is the anti-Stokes over Stokes ratio.

The derivative with respect to the temperature, has been evaluated at 514.5 nm, for all the Raman modes of anatase. The results, plotted in Figure 8b, show an increasing in sensitivity corresponding to the decreasing in frequency of the Raman modes and a decrease of the sensitivity of all Raman mode as temperature increases, also already observed in literature [43]. This can be attributed to the fact that at high temperature the differences between the population of the ground state and the first vibrational excited state are smaller than at room temperature.

In particular, it is possible to evaluate the sensitivity, at 300 K close to the room temperature, and to compare the obtained values for different wavelengths and different Raman modes of titanium dioxide. The outcome, shown in Table 6, is that the sensitivity at constant excitation wavelength decreases with increasing frequency of the Raman mode and for a given Raman mode it increases with increasing excitation wavelength. Corresponding to the 143 cm^−1^ E_g_ mode a thermal resolution in the range 1.2 (@ 514.5 nm) to 3.4 K (@ 647.1 nm) have been calculated. All these experimental data are comparable with those already reported in literature [9].

The behavior of the theoretical ratio can be compared to the experimental anti-Stokes/Stokes ratios for the four Raman modes, reported in Figure 6. It is evident that there is a good agreement, confirming the 143 cm^−1^ Raman mode, the lowest in Raman frequency, the more sensitive to Temperature variation. Moreover, the sensitivity calculated starting from the experimental data overlaps well the theoretical ones.

### 4.3. Validation of the Method and Temperature Determination: Test Sample

By using the calibration constants, it is possible to determine the local temperature. In Figure 9a, the temperature derived from repeated measurements, performed on a different region of the *test sample*, of the anti-Stokes/Stokes ratio for the 143 cm^−1^ mode of anatase at 514.5 nm, at 297 K, is reported; the figure also shows the mean temperature calculated for these measurements and the standard deviation. In this context, the standard deviation of temperatures, rather than the errors derived from each measurement, is reported, as it is preferable when repeated measures are conducted; moreover, the two values have the same order of magnitude (few kelvins). Detailed values are reported in Table 7, and it turns out that the data show an excellent overlap between the expected and measured data, with a deviation as low as 3 K, maximum. Moreover, the results obtained by changing the input power, reported in Figure 9b, confirm that the local sample temperature is not affected by the laser power, when an incident power of few mW is used, while at higher input powers the laser is heating the sample, at all the used wavelengths.

These results confirm the validation of the method thus verifying that the temperature measured all over the sample is uniform.

It is possible to conclude that the Raman modes of anatase, in particular the E_g_ one at 143 cm^−1^, are excellent candidate for the local temperature detection in the visible range. However, the need remains to investigate the behavior at longer wavelengths, towards the near IR, where the biological window is located.

The performances obtained with this TiO_2_ based Raman thermometer are compatible with data reported in literature [9,44]; the lower thermal resolution, with respect to fluorescent thermometer, is compensated with the wider wavelength working range.

## 5. Conclusions

Temperature is an important parameter influencing physical, chemical and biological processes: For this reason, the investigation of new materials, with enhanced performances, together with the definition of the more performing experimental protocol is a hot topic in the nanothermometry field. The experimental work reported in this article will contribute to the development of a new Raman based biocompatible nanothermometer, by investigating the optical performances of titanium dioxide, as anatase, with Raman technique.

The spectroscopic characterization of titanium dioxide has been carried out in the visible range, at 488.0, 514.5, 568.2 and 647.1 nm, and the Raman-active modes have been investigated to find the more performing one, as temperature sensor. Both Stokes and anti-Stokes spectra were collected at different temperature, input power and wavelengths, to investigate the temperature range, the temperature resolution, the eventual self-heating (due to the input laser power) of the sample and to identify the working range of the nanothermometer. A key point for the identification of the local temperature is the calibration of the experimental set-up, which allows defining the best experimental protocol. The calibration procedure has been conducted by controlling the sample temperature with a temperature-controlled stage and exploring the Raman signals in the temperature range of 283–323 K (with 5 K increment), as it is of interest for biological applications. The obtained values of the anti-Stokes/Stokes ratio allow the determination of the calibration constant, specified for all anatase Raman modes at each excitation wavelength. The calibration constant permits to determine the sample local temperature and to identify the power range where the local temperature is not affected by the laser power. Working with an incident laser power higher than 2 mW the sample experiences self-heating, while at lower power samples do not experience any self-heating. The validation of the proposed protocol has been finally achieved with the analysis of the Raman spectra of the *Test Sample*. Repeated anti-Stokes and Stokes measurements, have been performed on various positions of the sample at room temperature (~297 K), with an incident laser power of 1.5 mW. An excellent agreement between the temperature derived from the anti-Stokes/Stokes ratios and the expected temperature was found, with a standard deviation of repeated temperature measurements calculated to be in between 1 and 3 K, for the most intense peak, located at 143 cm^−1^, which has been demonstrated to be the most sensitive to temperature. This titanium dioxide mode seems to be an excellent candidate for the local temperature detection in the visible range from 488.0 to 647.1 nm, reaching the highest sensitivity in the red region.

## Figures and Tables

**Figure 1 biosensors-11-00102-f001:**
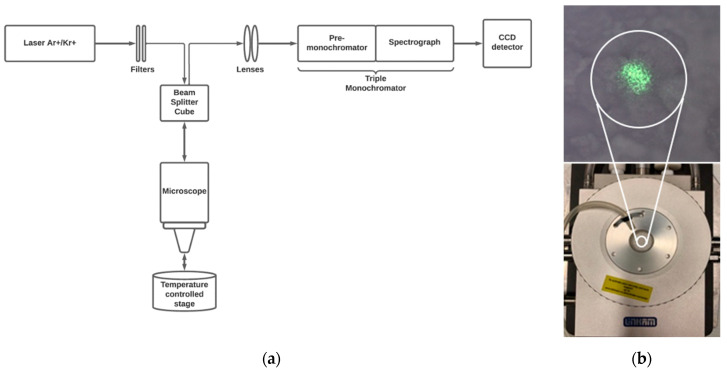
Experimental micro-Raman setup (**a**) and Linkam THMS600/720 temperature-controlled stage with zoom on the sample inserted, showing the laser spot (**b**).

**Figure 2 biosensors-11-00102-f002:**
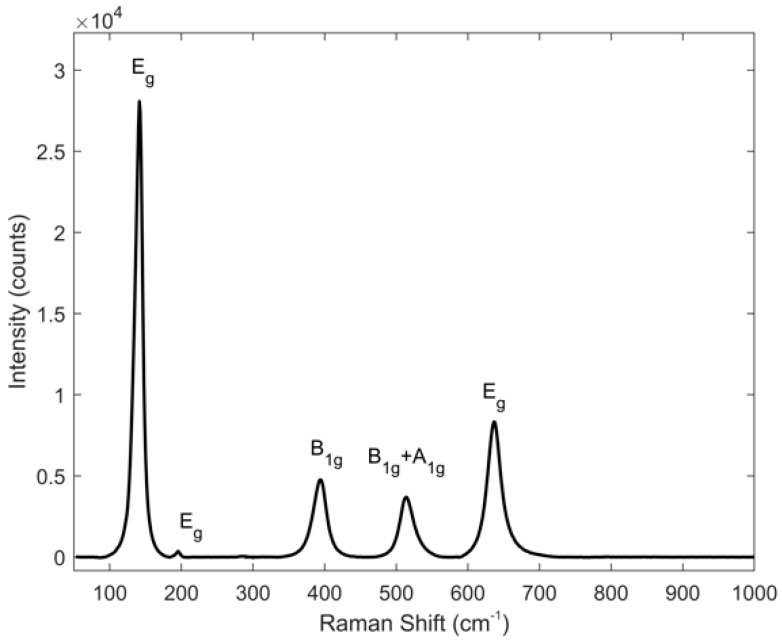
Stokes Raman spectrum of titanium dioxide anatase excited at 488.0 nm, with input power of 1.72 mW, with the indication of the Raman mode symmetry.

**Figure 3 biosensors-11-00102-f003:**
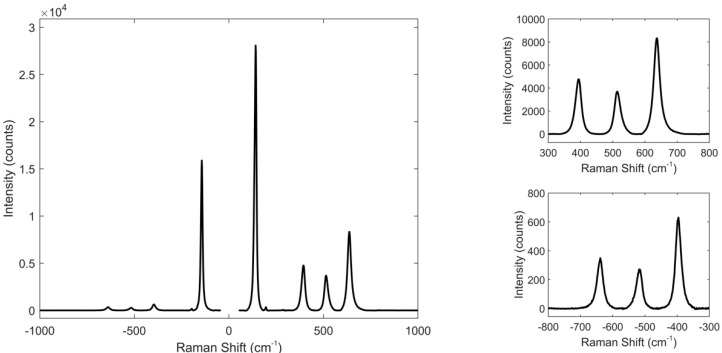
Stokes (positive Raman shift) and anti-Stokes (negative Raman shift) Raman spectrum of TiO_2_ anatase recorded at 488.0 nm, with input power of 1.6 mW. In the right panel the zooms of the three less intense peaks are displayed, Stokes in the upper part, anti-Stokes in the lower one.

**Figure 4 biosensors-11-00102-f004:**
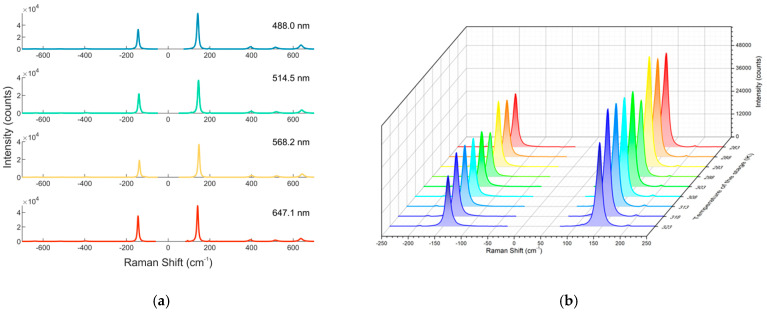
(**a**) Experimental anti-Stokes and Stokes spectra of TiO2 collected at room temperature and different excitation wavelengths (488.0 nm blue line, 514.5 nm green line, 568.2 nm yellow line and 647.1 nm red line); (**b**) Stokes and anti-Stokes spectra of the 143 cm^−1^ E_g_ mode, collected at 488.0 nm as function of temperature (from 283 to 323 K, with 5 K increment step).

**Figure 5 biosensors-11-00102-f005:**
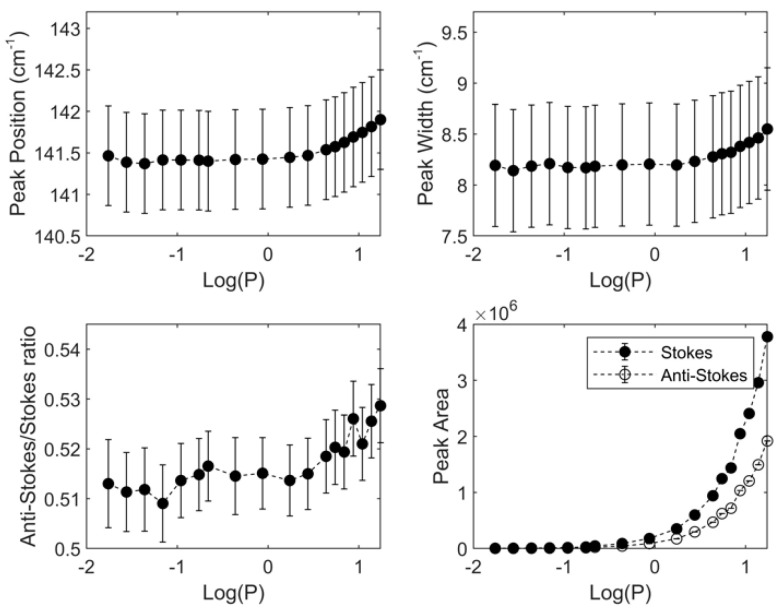
Peak position, peak width, anti-Stokes/ Stokes ratio and peak area for the 143 cm^−1^ E_g_ mode of anatase as function of the laser power incident on the sample; $\lambda_{exc}$ = 514.5 nm. The dashed lines are only for eyes guidance. All data are reported against the laser Power (mW) in a logarithmic scale.

**Figure 6 biosensors-11-00102-f006:**
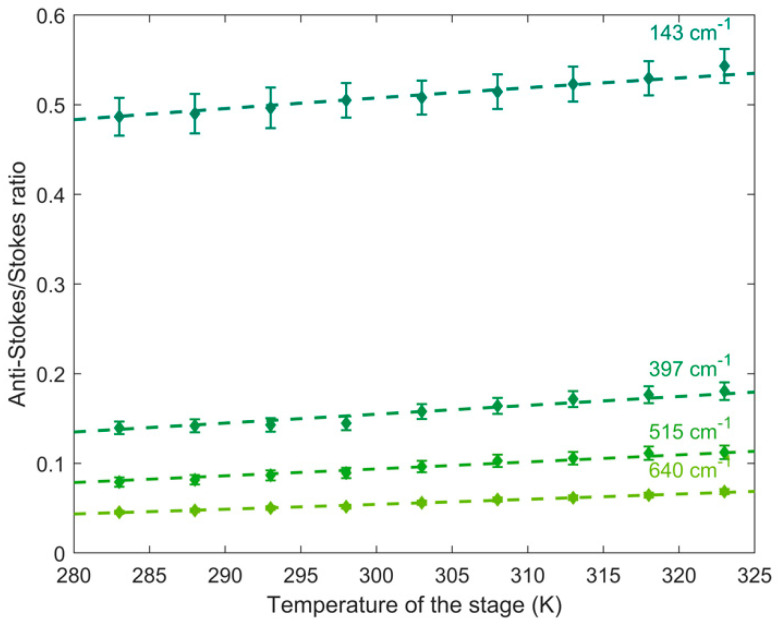
Experimental anti-Stokes/Stokes ratios in the range 283–323 K for the Raman modes of anatase, at 143 cm^−1^ (circles), 397 cm^−1^ (down-pointing triangles), 515 cm^−1^ (squares) and 640 cm^−1^ (up-pointing triangles) collected at $\lambda_{exc}$ = 514.5 nm.

**Figure 7 biosensors-11-00102-f007:**
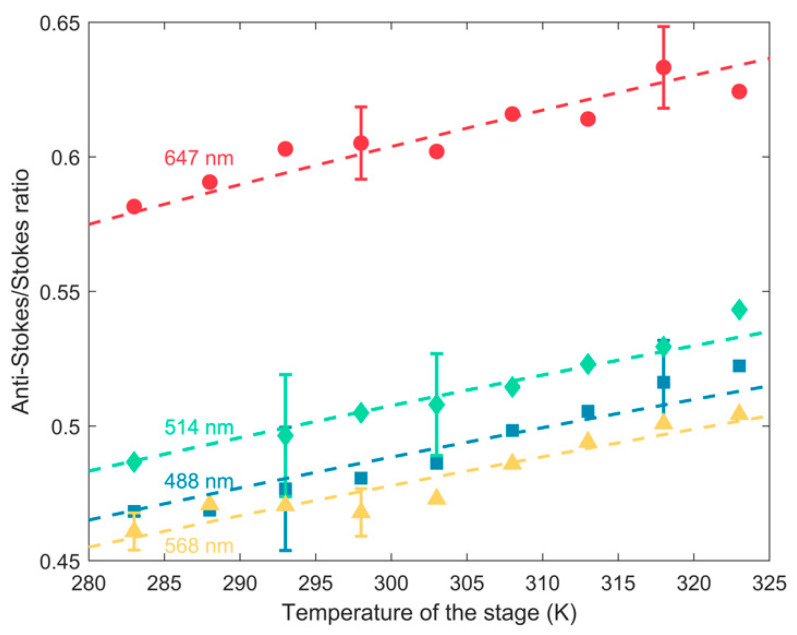
Experimental anti-Stokes/Stokes ratios of the 143 cm^−1^ E_g_ mode of anatase as a function of the temperature, and corresponding calibration curves; data are collected by exciting at different wavelengths: 488.0 (blue squares and dashed line), 514.5 (green diamonds and dashed line), 568.2 (yellow triangles and dashed line) and 647.1 nm (red circles and dashed line).

**Figure 8 biosensors-11-00102-f008:**
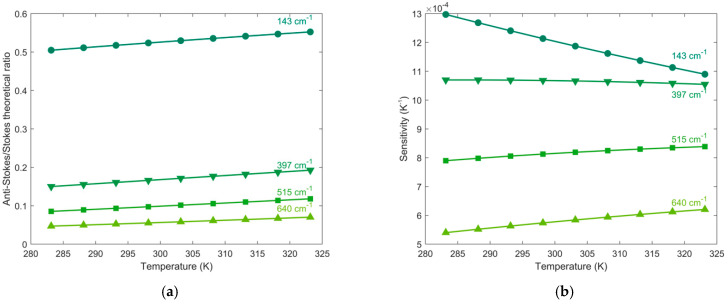
(**a**) Theoretical behavior and (**b**) sensitivity of the anti-Stokes/Stokes ratio of the four anatase modes, at 143, 397, 515, and 640 cm^−1^, calculated with $\lambda_{exc}$ = 514.5 nm, in the range 283–323 K.

**Figure 9 biosensors-11-00102-f009:**
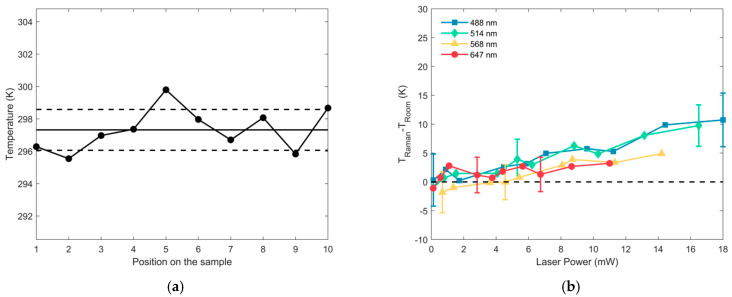
(**a**) Temperature determined from the anti-Stokes/Stokes values of the 143 cm^−1^ E_g_ mode at $\lambda_{exc}$ = 514.5 nm of the *Test Sample*, with the mean value (solid line) and the standard deviation (dashed lines); (**b**) difference between the temperature derived from the anti-Stokes/Stokes ratio (corrected with the proper calibration constant) for the 143 cm^−1^ E_g_ mode of anatase as function of the laser power incident on the sample at 488.0 (blue squares), 514.5 (green diamonds), 568.2 (yellow triangles) and 647.1 nm (red circles). The dashed line represents the room temperature.

**Table 1 biosensors-11-00102-t001:** Experimental and literature [28] TiO_2_ anatase Raman-active modes, excited at 514.5 nm.

Symmetry	Experimental [cm^−1^]	Literature [cm^−1^]
E_g_	143.0	143
E_g_	197.8	198
B_1g_	394.8	395
B_1g_	516.1	512
A_1g_	516.1	518
E_g_	638.4	639

**Table 2 biosensors-11-00102-t002:** Frequency position and width of the Stokes peak at 143 cm^−1^, Stokes and anti-Stokes areas and anti-Stokes/Stokes ratios for the temperature range 283–323 K, explored during the calibration procedure at 514.5 nm.

T [K]	Peak Center [cm^−1^]	Peak Width [cm^−1^]	$\mathrm{Stokes}\;\mathrm{Area}\;10^{5}$	$\mathrm{Anti}-\mathrm{Stokes}\;\mathrm{Area}\;10^{5}$	Anti-Stokes/Stokes Ratio
283	142.4 ± 0.6	10.7 ± 0.6	8.45 ± 0.32	4.11 ± 0.08	0.487 ± 0.021
288	142.3 ± 0.6	11.1 ± 0.6	9.29 ± 0.37	4.55 ± 0.09	0.490 ± 0.022
293	142.4 ± 0.6	11.4 ± 0.6	10.68 ± 0.44	5.30 ± 0.11	0.497 ± 0.023
298	143.0 ± 0.6	10.7 ± 0.6	7.00 ± 0.23	3.54 ± 0.06	0.505 ± 0.019
303	143.4 ± 0.6	10.8 ± 0.6	8.68 ± 0.29	4.41 ± 0.08	0.508 ± 0.019
308	143.5 ± 0.6	11.0 ± 0.6	9.16 ± 0.30	4.71 ± 0.08	0.515 ± 0.019
313	143.8 ± 0.6	11.0 ± 0.6	9.63 ± 0.32	5.04 ± 0.09	0.523 ± 0.020
318	144.3 ± 0.6	10.9 ± 0.6	8.99 ± 0.28	4.75 ± 0.08	0.530 ± 0.019
323	144.9 ± 0.6	10.8 ± 0.6	7.63 ± 0.23	4.15 ± 0.07	0.543 ± 0.019

**Table 3 biosensors-11-00102-t003:** Repeated measurements of the anti-Stokes/Stokes ratio at room temperature collected at 488.0, 514.5, 568.2 and 647.1 nm using a laser power of 1–2 mW depending on the excitation wavelength used. In the final row, mean values of the anti-Stokes/Stokes ratios and the standard deviations are reported.

Anti-Stokes/Stokes Experimental Ratio				
Measurements	@ 488.0 nm	@ 514.5 nm	@ 568.2 nm	@ 647.1 nm
1	0.490 ± 0.010	0.5026 ± 0.007	0.477 ± 0.006	0.609 ± 0.008
2	0.486 ± 0.009	0.5012 ± 0.007	0.471 ± 0.006	0.602 ± 0.008
3	0.487 ± 0.010	0.5028 ± 0.007	0.477 ± 0.006	0.610 ± 0.008
4	0.483 ± 0.009	0.5030 ± 0.008	0.474 ± 0.006	0.609 ± 0.007
5	0.489 ± 0.009	0.5058 ± 0.007	0.478 ± 0.006	0.607 ± 0.007
6	0.485 ± 0.009	0.5034 ± 0.007	0.471 ± 0.006	0.597 ± 0.008
7	0.488 ± 0.009	0.5017 ± 0.007	0.476 ± 0.006	0.609 ± 0.008
8	0.486 ± 0.009	0.5034 ± 0.007	0.474 ± 0.006	0.605 ± 0.007
9	0.488 ± 0.009	0.5006 ± 0.007	0.477 ± 0.006	0.608 ± 0.007
10	0.486 ± 0.009	0.5040 ± 0.007	0.474 ± 0.006	0.606 ± 0.007
Mean ± σ	0.487 ± 0.002	0.503 ± 0.001	0.475 ± 0.002	0.606 ± 0.004

**Table 4 biosensors-11-00102-t004:** Anti-Stokes/Stokes experimental ratios at different excitation wavelengths, as function of the temperature imposed by the thermostat.

Anti-Stokes/Stokes Experimental Ratio vs. Temperature				
T [K]	@ 488.0 nm	@ 514.5 nm	@ 568.2 nm	@ 647.1 nm
283	0.468 ± 0.023	0.487 ± 0.021	0.461 ± 0.007	0.582 ± 0.012
288	0.469 ± 0.023	0.490 ± 0.022	0.471 ± 0.008	0.591 ± 0.014
293	0.477 ± 0.023	0.497 ± 0.023	0.471 ± 0.008	0.603 ± 0.014
298	0.481 ± 0.022	0.505 ± 0.019	0.468 ± 0.009	0.605 ± 0.013
303	0.486 ± 0.021	0.508 ± 0.019	0.473 ± 0.009	0.602 ± 0.015
308	0.498 ± 0.020	0.515 ± 0.019	0.486 ± 0.009	0.616 ± 0.015
313	0.506 ± 0.018	0.523 ± 0.020	0.494 ± 0.008	0.614 ± 0.015
318	0.516 ± 0.016	0.530 ± 0.019	0.501 ± 0.008	0.633 ± 0.015
323	0.522 ± 0.016	0.543 ± 0.019	0.504 ± 0.008	0.624 ± 0.015

**Table 5 biosensors-11-00102-t005:** Calibration constants for the 143 cm^−1^ Raman mode at four different wavelengths.

Excitation Wavelength [nm]	Calibration Constant	R^2^
488.0	0.931 ± 0.009	0.91
514.5	0.961 ± 0.006	0.95
568.2	0.904 ± 0.007	0.90
647.1	1.135 ± 0.009	0.84

**Table 6 biosensors-11-00102-t006:** Sensitivity at 300 K of the anti-Stokes/Stokes ratio to temperature, calculated for the 143, 397, 515 and 640 cm^−1^ modes of anatase.

$\mathrm{Sensitivity}\;\mathrm{at}\;300\;K\;[10^{-3}\;K^{-1}]$				
Excitation Wavelength [nm]	Raman Mode			
	143 cm^−1^	397 cm^−1^	515 cm^−1^	640 cm^−1^
488.0	1.20	1.06	0.81	0.57
514.5	1.20	1.07	0.81	0.57
568.2	1.22	1.08	0.83	0.59
647.1	1.23	1.10	0.85	0.60

**Table 7 biosensors-11-00102-t007:** Results of repeated measurements at four different excitation wavelengths conducted at room temperature.

Excitation Wavelength [nm]	Laser Power [mW]	Room Temperature [K]	Averaged Raman Temperature [K]	Root Mean Square Deviation [K]
488.0	1.72	297.0	298	2
514.5	1.52	297.0	297	1
568.2	1.35	296.0	296	2
647.1	1.09	298.0	299	3

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
