# Peer review of "Contactless Temperature Sensing at the Microscale Based on Titanium Dioxide Raman Thermometry"

_biosensors, 2021, doi:10.3390/bios11040102_

Round 1

Reviewer 1 Report

In this manuscript, local temperatures are derived from ratios of anti-Stokes to Stokes Raman from TiO2 particles. The authors investigated the effect of excitation wavelength, power, and Raman mode on the derivation of the temperature. The temperature dependence of the ratio is influenced by the different response of the experimental setup to the wavelength. The information is important for the nanothermometry. Thus, this may be publishable in the journal. 

However, there are minor questions and problems as follows. 

Line 84: Does Raman signal depend on the 4th power of the excitation frequency, rather than the 3rd power of that?

Fig. 6 and 7 show the temperature dependence by excitation under low laser power. The results also under the high power can be displayed. I am wondering whether the slope (the calibration constant) is changed. 

Fig. 8 shows the theoretical results, and Fig. 8a can be compared with the experimental result as shown in Fig. 6. To compare with Fig. 8b, the experimental sensitivity derived from Fig. 6 should be expressed. 

Ref. 18: Information of the journal, volume, and page are needed.  

Ref. 19: Information of the page is missing. 

Ref. 34: Information of the year is missing. 

Author Response

We thank the reviewer 1 for his positive comments on our paper and his suggestions. We have corrected the citations outlined (reffs 18, 19 and 34). The responses on minor questions and problems, outlined by referee, are reported in the following:

  1. Line 84: Does Raman signal depend on the 4th power of the excitation frequency, rather than the 3rd power of that?” As outlined by reviewer we have reported on line 84 “… the Raman signal depends on the third power of the excitation frequency…”. We have corrected this statement by including also a forth power dependence (see text). The motivation follows. Different papers already published in literature report a dependence on the third or on the forth order of the power of the excitation frequency. Ref 39, clarifies this point by distinguishing between the different signal acquisition methods, underlined in our paper below, lines 237-240: “A frequency dependence to the third power of the anti-Stokes/Stokes ratio is needed as the detection system is based on photon counting (CCD), whereas if the detection is energy based a fourth power dependence is more appropriate [39]”. Since in our experimental set-up we are using a CCD detector, we should consider a dependence on the third power.
  2. Fig. 6 and 7 show the temperature dependence by excitation under low laser power. The results also under the high power can be displayed. I am wondering whether the slope (the calibration constant) is changed”. The goal of the present work is the detection of the local temperature without altering it with a laser source, to this end only by working at low laser power (see results of figure 5) we are able to obtain reliable results. The effect of increasing the laser power, already visible in figures 5 and 9, will be the subject of further measurements to be properly performed.
  3. Fig. 8 shows the theoretical results, and Fig. 8a can be compared with the experimental result as shown in Fig. 6. To compare with Fig. 8b, the experimental sensitivity derived from Fig. 6 should be expressed”. As suggested by referee we have calculated also the experimental sensitivity, from data reported in figure 6, the results reported below, are very similar to the theoretical ones. We have added a comment on the paper, without reporting the picture, since it does not provide further remarkable information.

Reviewer 2 Report

In the present manuscript, the authors report the results of an investigation on the use of Raman spectroscopy for Titanium Dioxide temperature measurements by means of the Stokes and anti-Stokes intensity ratio. The manuscript certainly presents some interesting aspects but some parts are not treated with the necessary attention. The abstract should be shortened and revised in order to be more focused on the experimental results really reported in the manuscript. Also in the introduction, the authors should clarify the aims of the present paper, many points cited in the introduction are not adequately addressed in the manuscript. The possibility to use the propose methid in biologiacal samples is actually not discussed in details, so it should be made as a minor ppoint of the abstract and the introduction. In the Materials and Methods section, some important details as the spectral resolution of Raman spectrometer or details on the fitting procedure are missed. In the Results section, quite all the data are reported without any attention to experimental errors. A procedure for evaluating a method for quantifing the temperature should involve the use of a thermometer whose output can be considered as a reference vaue. This is not the case. It should bebetter discussed. For these reasons, the paper can not be accepted in the present form and has to be deeply revised.

A more detailed list of the points to be addressed by the authors is reported in the following.

-Please check lines 64-65, it seems that some words are missing. In statement b), please list also eventual changes in the samples or processes eventually triggered by the resonant absorption

- Please indicate the spatial resolution of the Raman spectrometer and the precision in the temperature measurements in the controlled-temperature stage. Please add information about the positions of the samples where the the sample test measurements were taken (always in the same point or from different positions?)

- At lines 140-141, the authors say that “Before starting the experiment, a procedure of purging air from the stage chamber with nitrogen is performed”. Please explain the reason for performing this procedure.

- Lines: 146-147: What does "a consistent set of data " mean?

- In Tables 1 and 2 the Raman modes are reported as integer or with a decimal digit. Why there is this difference? How the authors can distinguish mode positions that differ for less than 1 cm-1 and they are not able to separate the two contributions at 512 and 518 cm-1 in the TiO2 Raman spectrum (line 168)?

- Please add errors in the values of the peak areas and ratios in Tables 2 and 3, and in related Figures.

- Please explain in a more exhaustive way the choice of Equation 1 by consulting the references reported by Tuschel (ref. 39 of the present manuscript).

- Please remove the line connecting the points shown in the upper panels of Fig.5.

- The reported fitted functions related to eq. 1 seem to be quite similar to alinear behaviour in the investigated cases. Please add details about the fitting procedures (i.e., details about parameters for evaluating the goodness of the fit).

- Please add errors also in Tables 4,5 and 6 and in Figures 6, 7, 8.

- As far as concerns 4.3 section, please revise the title. The results reported in Figure 9b should be better explained. In order to validate the method also changes in temperature should be considered.

- Please carefully revise the English language.

-Please check the style of some references

Author Response

The authors are highly thankful to the reviewer for all the remarks on the present article. We have revised the article as per the comments/suggestions, as outlined on the paper.

The attached file contains a detailed response.

Reviewer 3 Report

In this manuscript, the authors propose a method for temperature sensing at the nanoscale. While their study should be of high interest to many potential applications, it fails to meet the expectation of the reader. I would strongly suggest significant improvement in experimental designs, analysis and proper proof of significance before I can consider it for publication as an article in Biosensor. Here are some of my key concerns:  

  1. What do the authors mean by temperature sensing at “Nanoscale”? Can their system detect temperature variations of nano-kelvin? Or do their system measure temperature variations between two positions distanced by few nanometers? Clarifications are warranted.
  2. All the datasets lack error bars. That makes the study unreliable. One can argue, that given the relatively small change in the AS/S ratio as a function of temperature, the errors are very relevant and can determine the validity of the study.
  3. Along the same line, is it relevant to quote four significant digits after the decimal point? This must be justified by the error estimation.
  4. What is the difference between Figure2 and Figure3? Figure 2 seems to be the subset of Figure-3!
  5. Line :257, “The corresponding calibration constants are calculated to be 0.9605, 0.9411, 0.9461, 0.9491, respectively.” What is the significance of these constants? Rather, do the authors feel that a linear fit in this temperature range would be more informative, where one does not need to know the exact value of the laser excitation lines, position of the AS and S lines? One can find a linear (close enough) relationship?
  6. Table-2: can the authors describe why, despite of using the same laser power and all other settings, the absolute value of the AS and S lines change vigorously? Can the authors determine whether the ratio change arise from increase in the AS are, or decrease of the S-area or combination of both?
  7. Line-203: “It is evident that parameters are not increasing with the laser power only in the small range, 0-2 mW.” Is this justified? Are the variations not pretty linear throughout the power range?
  8. Table3: this represents the low sensitivity of their method! One would expect a variation of temperature gradient throughout different portion of the sample. For example, the portions touching the heater should have a higher temperature, compared to the central part where the objective/ air works as a sink? If the authors claim a sensitive thermometer, one would expect to see a temperature gradient profile as a function of sample chamber position.
  9. Table7: The authors claim to have a RMS deviation of 1-3K. for 10 different temperature measurements. Why is this better compared to any commercial handheld UV-thermometer for example? The authors should clarify why their complicated thermometer better than any other alternatives?

Author Response

The authors are highly thankful to the reviewer for all the remarks on the present article. We have revised the article considering all suggestions. Below is the required explanation on some important points outlined by referee 3:

  1. What do the authors mean by temperature sensing at “Nanoscale”? Can their system detect temperature variations of nano-kelvin? Or do their system measure temperature variations between two positions distanced by few nanometers? Clarifications are warranted.”

Temperature sensing at the nanoscale means that the temperature can be measured with sufficient precision and with a high spatial resolution, reaching the nanometric scale. The spatial resolution, working with micro-Raman set-up, is strictly related to the diffraction limit of the used objective, affecting the spot diameter that could be close to lexcitation /2. When lexcitation=514.5 nm, the spot diameter, using a 100x objective lens, could be less than 300 nm.

The combination of this set-up with active materials with nano-scale dimensions will open the way to reach sensing at the nanoscale, which will be essential for future application in the biological field.

  1. All the datasets lack error bars. That makes the study unreliable. One can argue, that given the relatively small change in the AS/S ratio as a function of temperature, the errors are very relevant and can determine the validity of the study” and “Along the same line, is it relevant to quote four significant digits after the decimal point? This must be justified by the error estimation”.

We completely agree with this observation. We have added on tables and figures all the error bars, together with the explanation on how they have been calculated, and corrected the number of the significant digits according to the errors, to demonstrate the reliability of all the presented data and the validity of our work.

  1. What is the difference between Figure2 and Figure3? Figure 2 seems to be the subset of Figure-3!”.

In Figure 2 we have reported only the Stokes part of the Raman spectrum of TiO2, by including information on the symmetry of the Raman modes. Figure 3 reports both Stokes and anti-Stokes peaks and the zoom on the three small intense peaks, that are not well visible in the figure. We have divided data in two different figures in order to make easier and direct the information and the comparison with published data, reporting in general only the Stokes Raman spectrum. If necessary we can join the two figures and label the Stokes peaks to link the figure to table.

  1. Line :257, “The corresponding calibration constants are calculated to be 0.9605, 0.9411, 0.9461, 0.9491, respectively.” What is the significance of these constants? Rather, do the authors feel that a linear fit in this temperature range would be more informative, where one does not need to know the exact value of the laser excitation lines, position of the AS and S lines? One can find a linear (close enough) relationship?

The calibration constant is determined by measuring the AS/S ratios at different temperatures and then fitting the experimental data with the equation (1). The nature of the dependence of the AS/S ratio on T is an exponential one (Boltzmann), not a linear one, so a linear fitting would be misleading and unrelated to the physical reality of the underlying phenomenon. These important observations have been reported also in appendix A, properly prepared, where details and meanings of the fitting are explained. A fitting considering a linear dependence will give out a constant value around 0.57, which give final temperature values very far from the ones obtained with the constant calculated taking into account the exponential behaviour.

  1. Table-2: can the authors describe why, despite of using the same laser power and all other settings, the absolute value of the AS and S lines change vigorously? Can the authors determine whether the ratio change arise from increase in the AS are, or decrease of the S-area or combination of both?

Measurements described in table 2 are conducted at the same laser power but with varying temperature. The AS/S ratio varies with temperature as the population of the first vibrational state of the electronic ground state increases with increasing temperature. The ratio changes as a combination of two factors: the Stokes area decreases as it depends on the population of the ground state, while the anti-Stokes area increases as it depends on the population of the first vibrational level. Also these important observations have been reported in appendix A.

  1. Line-203: “It is evident that parameters are not increasing with the laser power only in the small range, 0-2 mW.” Is this justified? Are the variations not pretty linear throughout the power range?

To convince referee and any reader we have performed new measurement at low input power and inserted these new measurements in the revised version of the paper. Data are reported in a logarithmic scale in order to evidence two regions: at low input power (in between 0-2 mW) the temperature is clearly (more then with the previously reported measurements) constant, while at higher values the temperature is increasing with the increase of the input power, thus indicating a local heating induced by the input laser.

  1. Table3: this represents the low sensitivity of their method! One would expect a variation of temperature gradient throughout different portion of the sample. For example, the portions touching the heater should have a higher temperature, compared to the central part where the objective/ air works as a sink? If the authors claim a sensitive thermometer, one would expect to see a temperature gradient profile as a function of sample chamber position.

This is a good observation when solid plates are used to heat a sample! We have added information of our temperature controller, which will help to unravel this doubt! Our temperature controller heats and cools down the sample with nitrogen gas, whose temperature is finely controlled and so thermalizes samples. By this way, there is not a temperature gradient in the sample so it is not detected by our temperature measurement. The sample is at a uniform temperature and the temperature is measured in the exact point in which the laser is focused on the sample. We irradiate the sample with the laser and we collect the scattering of the same laser spot, which is linked to temperature.

  1. Table7: The authors claim to have a RMS deviation of 1-3K. for 10 different temperature measurements. Why is this better compared to any commercial handheld UV-thermometer for example? The authors should clarify why their complicated thermometer better than any other alternatives?” Commercial handheld UV-thermometers are IR Thermometer with UV Leak Detector, they work with high precision and sensitivity by detecting the temperature on samples with large area, dimensions of few cm or mm. Our complicated thermometer allows determining temperature on micro-scaled objects, using nano-sized materials. This work is presented as a proof of principle indicating the usefulness of anatase and micro-Raman as material and technique for temperature detection. Further studies will be pursued to improve the response of this sensor, by synthetizing new titania based materials and enhancing Raman efficiency. A final consideration can be done by considering many favourable attributes of this technique: small volumes, adaptability to cw or pulsed modes of sample irradiation, detection on gas, liquid and solid samples, and high spatial resolution, which is higher for Raman spectroscopy than for IR thermometry (see reffs. Carlos D. S. Brites, et al., Thermometry at the nanoscale, Nanoscale, 2012, 4, 4799; Marta Quintanilla and Luis M. Liz-Marzán, Guiding Rules for Selecting a Nanothermometer, Nano Today, 2018, 19, 126-145).

Round 2

Reviewer 2 Report

Quite all the points have been addressed. It is still not fully clear to the reader the question related to the point
1. “How the authors can distinguish mode positions that differ for less than 1 cm-1 and they are not able to
separate the two contributions at 512 and 518 cm-1 in the TiO2 Raman spectrum (lines 172-174)?”

The  authors have clearly explained the point in the answer to the referees. It should be provided also in the manuscript,ootherwise statement of lines 172-174 can be misleading.

Author Response

We thank referee for the response. We have revised the article by providing an explanation of statement of lines 172-174, as required.

Reviewer 3 Report

While the authors made significant changes and tried to answer some of my concerns, it needs more clarification before I can recommend for publication. Here is a list of my concerns.

  1. The heading is clearly claiming beyond the paper can scientifically achieve. The theoretical estimation of lamda/2, Rayleigh criterion of spatial resolution does not in any way allow their method to be nano-scale sensitive. Unless they have a physical proof that they can measure temperature variations within whatever limit they say, I would not refer it as a "nanoscale" temperature measuring device.
  2.   The authors should have a section explaining the error calculation. Why is all the errors of peak position and peak width same? How did the authors calculate error propagation? A quick check of some of the AS/S ratio errors did not agree with my calculations.
  3. In table 2 the AS area and the S area do not follow any particular trend and that is my concern. The authors should explain clearly, why they think, the AS area increases and the S area decreases with temperature?
  4. Figure5 makes my point about error estimation more relevant, as you can see with the error bars, the change is marginal with temperature. I am thus not convinced that this is a sensitive way to measure temperature. The authors should at least comment on the shortcomings of the study in the discussion.

Author Response

The authors are highly thankful to the reviewer for all the considerations on the present article.

According to the observations, we have revised the article by adding new remarks, as outlined in the attached file.
